# Ultrasound Imaging of Acquired Myometrial Pseudoaneurysm: The Role of Manipulators as an Unusual Cause during Laparoscopic Surgery

**DOI:** 10.3390/diagnostics12010164

**Published:** 2022-01-11

**Authors:** Francesca Buonomo, Clarice de Almeida Fiorillo, Danilo Oliveira de Souza, Fabio Pozzi Mucelli, Stefania Biffi, Federico Romano, Giovanni Di Lorenzo, Sofia Bussolaro, Giuseppe Ricci

**Affiliations:** 1Institute for Maternal and Child Health, I.R.C.C.S. Burlo Garofolo, 34137 Trieste, Italy; clarice.dealmeidafiorillo@burlo.trieste.it (C.d.A.F.); stefania.biffi@burlo.trieste.it (S.B.); federico.romano@burlo.trieste.it (F.R.); giovanni.dilorenzo@burlo.trieste.it (G.D.L.); giuseppe.ricci@burlo.trieste.it (G.R.); 2ELETTRA Sincrotrone Trieste S.C.p.A., S.S. 14 Km 163.5, Basovizza, 34149 Trieste, Italy; danilo.oliveiradesouza@elettra.eu; 3Department of Diagnostic and Interventional Radiology, Cattinara Hospital, 34149 Trieste, Italy; fabio.pozzimucelli@aots.sanita.fvg.it; 4Department of Medical, Surgical and Health Science, University of Trieste, 34127 Trieste, Italy; sofia.bussolaro@burlo.trieste.it

**Keywords:** uterine artery pseudoaneurysm, ultrasound, dimension, computed tomography, magnetic resonance

## Abstract

An acquired uterine artery myometrial pseudoaneurysm can occur due to inflammation, trauma, or iatrogenic causes, such as surgical procedures, and can lead to profuse bleeding. The efficacy of uterine manipulators in gynecological surgery, particularly as a cause of a pseudoaneurysm, has been poorly discussed in the literature. In this paper, we discuss a case of a 39-year-old woman with profuse uterine bleeding that occurred seven days after operative laparoscopic surgery for endometriosis. The color Doppler ultrasound better evoked the arterial-like turbulent blood flow inside this cavity. These sonographic features were highly suggestive of uterine artery pseudoaneurysm, presumably related to a secondary trauma caused by the manipulator. The diagnosis was subsequently re-confirmed by angiography, and the patient was treated conservatively with uterine artery embolization. Ultrasound has been shown to be a valuable and safe tool for imaging pseudoaneurysm and guiding subsequent interventional procedures. Accordingly, we briefly review the most suitable manipulators used in benign gynecological surgeries to verify if the different types in use can guide the surgeon towards the correct choice according to surgical needs and thus prevent potentially dangerous trauma.

## 1. Introduction

Acquired uterine artery pseudoaneurysm (UAP) is an intramyometrial blood-filled cavity communicating with an arterial vessel with the surrounding perivascular tissue creating the pseudoaneurysm wall [1]. Acquired pseudoaneurysms differ from true aneurysms because they are not surrounded by the three arterial wall layers of the arterial wall (intima, media, and adventitia). Instead, they usually contain a single layer of loose connective tissue, and this extra-luminal blood flow can enlarge and rupture [2]. The predominant symptom of a UAP consists of intermittent vaginal bleeding with life-threatening hemorrhage [3], which can be prevented by early detection and therapeutic intervention [4]. UAP may arise as an outcome of a rupture in the continuity of the arterial wall due to trauma, inflammation, tumor, or iatrogenic causes, such as surgical procedure, drainage, or percutaneous biopsy [5]. Moreover, surgery can lead to pseudoaneurysm formation through direct injury or infection of one or all the layers of the vessel [6]. It usually occurs after cesarean delivery [7], after spontaneous or operative vaginal delivery and surgical abortion [1], as well as being a possible fatal cause of abdominal pain in a pregnant woman who had a uterine pseudoaneurysm due to deep endometriosis [8]. In the literature, it is further reported after gynecologic surgical procedures such as laparotomic, laparoscopic, and hysteroscopic myomectomy, excision of deep endometriotic lesions, conization of the uterine cervix [9], and other rare diseases, such as spontaneous thrombosis [10,11] and inflammatory conditions [12,13].

The uterine manipulator, a tool widely used to allow mobilization of the uterus by exposing the operating field, is a surgical device that has been associated with the onset of complications [14]. Nevertheless, its use is essential in some types of gynecological surgery because it reduces operating time, prevents damage to the organs of the urinary tract [14], and promotes proper delineation of the vaginal fornices for colpotomy [15,16].

In this article, we describe a case of acquired UAP that developed after surgery for severe endometriosis. The uterine manipulator utilized for surgical site exposure is presumed to have caused myometrial damage, leading to the formation of a uterine pseudoaneurysm that was not present before surgery. In this context, ultrasound (US) diagnosis has been crucial for patient management and the prevention of further complications. Moreover, we review and discuss the conscious and careful use of the uterine manipulator and the appropriateness of each type used according to the surgery plan.

## 2. Case Report

A 39-year-old woman, gravida 3, para 1, underwent abdominal operative laparoscopy for infertility and dysmenorrhea due to endometriosis. The preoperative US of the pelvis (Figure 1A) was performed with Voluson E10 US system (GE Healthcare, Zipf, Austria) using a wideband 5–9-MHz endocavitary transducer and conducted according to IDEA (international deep endometriosis analysis) consensus [17] by a gynecologist ultrasound specialist with more than 25 years of expertise.

The US images, taken and saved electronically, revealed: (i)The presence of a hypoechoic, presumably endometriotic, nodule of approximately 2 cm in the fixed vesicouterine plica;(ii)Asymmetric myometrium and some minimal focal vacuolar areas in the anterior part of the uterine corpus, suggesting diffuse adenomyosis;(iii)A unilocular avascular ground glass cyst of (49 × 36 × 56) mm suggestive of a typical endometrioma in the left ovary;(iv)Both right and left uterosacral ligaments were thickened without lesions evident at the US.

Before laparoscopic surgery, after disinfection of the entire operating field, a Foley catheter was introduced. Then a non-reusable manipulator (ClearView^®^ Uterine Manipulator, 7 cm tip, Clinical Innovations, LLC, Murray, UT, USA) was inserted into the uterus through the vagina. The abdominal cavity inspection showed a diffusely increased uterine body volume, which was suggestive of adenomyosis. The left ovary had a cystic formation of about 5 cm attached to the ovarian fossa; meanwhile, the right ovary and the bilateral tubes were regular. Further, a 2 cm nodule on the vesico-uterine plica, a thickening of both uterosacral ligaments, and another 1 cm nodule on the right lateral paracolic gutter were detected. We performed radical asportation of pelvic endometriosis stage III (according to ASRM classification) at the level of the uterosacral ligaments and the pelvic nodule on the vesico-uterine plica and the paracolic gutter. We enucleated the endometriotic cyst and executed a chromosalpingography with the bilateral passage with methylene blue contrast medium. Once the endometriotic cyst was enucleated, the nodules were removed together with the radical asportation of pelvic endometriosis at the level of bilateral uterosacral ligaments. There was no significant blood loss during the surgery. The pathologic exam confirmed endometriosis in all samples received for analysis.

One day after the surgery, the patient was discharged in good health after a postoperative US (Figure 1B), where no abnormal features were detected. One week later, she complained of profuse vaginal bleeding. She had no fever, smelly vaginal discharge, or pelvic pain. Blood tests were regular at admission, with haemoglobin at the standard limit (12 g/dl) and a regular coagulation test. BETA-hCG (BhCG) presented values inside the normal range (negative). A second-level US was performed by the same specialist that carried out the “IDEA” preoperative evaluation. The transvaginal 2D US showed a rounded uterine intramyometrial hole with rapidly moving hypoechoic flow inside, measuring (1.6 × 1.5 × 1.5) cm, in the left part of the intrauterine wall. This image, not present in the preoperative US evaluation, confirmed the acquired origin of the lesion. The color Doppler US of this area allowed observation of the central arterial-like turbulent low impedance blood flow deriving from the ascending branch of the left uterine artery pointing to an acquired uterine pseudoaneurysm (Figure 2). In addition, a thin layer of myometrium left from the upper part of the uterine serosa was noted. Careful observation enabled verification of the distinctive “to-and-fro” waveform at the neck of the pseudoaneurysm. The observation of these typical features in the images provided certainty of the diagnosis and avoided further investigations. Considering the profuse blood loss, we informed the patient about the high risk of uterine rupture due to the post-surgical lesion. As a result of these findings, she underwent an angiographic evaluation for conservative embolization treatment.

The pelvic angiography, performed through retrograde left femoral access and catheterization of the left hypogastric artery, identified the left internal iliac artery and the distal branch of the ascending left uterine artery (Figure 3A). A nearby intramyometrial tubuliform dilatation appeared with a slightly smaller diameter and a different morphology than the previous ultrasound image. A contiguous intramyometrial contrast media diffusion presumably occurred shortly before the arteriography was performed as a possible expression of initial extravasation due to a probable rupture with blood transfer. Fortunately, the rupture zone occurred medially to the lesion in the intramyometrial zone and not towards the uterine serosa, where the lesion was very close. Otherwise, if the blood found a way out without resistance, the bleeding could be life-threatening.

A super-selective catheterization was performed with a microcatheter, and embolization of the left uterine artery was performed by releasing embolic polyethylene glycol particles (Hydropearl-Terumo 800 ± 75 μm). The subsequent control showed a very slow flow in the uterine artery and the disappearance of the pathological finding on the post-surgical report (Figure 3B). Afterwards, selective catheterization of the right hypogastric artery was conducted. Since no vascular alteration was seen in this uterine site, bilateral embolization was not necessary. There were no complications during the procedure, and then the bleeding ceased.

Three days after the angiographic procedure, the patient was in good condition and underwent a new US control. The lesion was still present with the same size, internally characterized by indistinct fixed echoes suggestive of the presence of residual blood, but without internal flow, which demonstrated that the embolization performed was adequate.

One week later, the complete resolution of the clinical situation was verified and confirmed by US. The patient had no abdominal pain or blood loss, and the patient’s blood tests were within the normal range. After this US control, she was discharged from the hospital. One week later, the patient underwent a new US, which confirmed a regular intrauterine wall.

## 3. Discussion

This report describes an original case of uterine artery pseudoaneurysm that happened after abdominal operative laparoscopy for endometriosis occurring one week after surgery, presumably caused by a uterine manipulator myometrial injury. To our knowledge, this is the second case described in the literature (besides ref. [4]).

An early and accurate diagnosis of UAP was performed using the color Doppler 2D US. Its high sensitivity in revealing the typical blood flow of the pseudoaneurysm, i.e., swirling within an anechoic sac-like structure [1,18,19], has demonstrated that color Doppler US is a valuable and safe tool for the diagnosis of pseudoaneurysms. Although such a method has diagnostic limitations, the operator’s clinical expertise in recognizing the typical features in the US may overcome or drastically minimize the issues arising from uterine artery bleeding.

Angiography is considered the “gold standard” technique for definitive diagnosis and treatment. As its main advantage, it has real-time evaluation capacity during hemodynamic monitoring [6] and provides a diagnostic tool with concomitant therapeutic potential, where indicated [20,21]. On the other hand, it is naturally invasive and presents real risks of complications associated with the procedure at the access site, such as hematomas or pseudoaneurysms, or at the level of the “target” lesion with embolization of “not target” vessels [20,22]. Additionally, it is an expansive image modality, and the requirement for ionizing radiation must be considered [23].

The computed tomography (CT) angiography has advantages over other imaging methods, including the US, magnetic resonance (MR) imaging, and angiography. It is not operator-dependent, and the isotropic volumetric data can be managed with 3D volume rendering [24,25,26]. Further, it has high sensitivity and specificity for detecting arterial injuries [20]. However, CT angiography is not a therapeutic tool, and angiography is required after the examination.

MR imaging techniques use less nephrotoxic contrast agents and no ionizing radiation [27]. While not the first imaging modality of choice for evaluating suspected UAP, 3D contrast angiography is helpful in imaging pseudoaneurysms in patients with allergies or impaired renal function to CT contrast material [23,28].

Currently, diagnostic imaging modalities comprise color Doppler US [29], 3D color/power Doppler US [30,31], CT, and 3D-CT [6,32]. Previously, the US was used as a screening tool, but several sources have shown that it can be employed as a diagnostic modality [29]. The characteristic to-and-fro pattern is a typical color Doppler US sign that has a sensitivity of nearly 95% [33,34]. It assesses the communicating channel (neck) between the sac and the feeding artery with a “to-and-fro” wave shape, the “to” representing the blood within the pseudoaneurysm during systole and the “fro” during diastole [35]. Color Doppler allows visualization of the swirling blood flow on this structure, with a typical “yin-yang” sign [36] (see the video in Appendix A). Nonetheless, the advantage of 3D reconstruction of color/power Doppler US images can enable us to understand the vessels’ spatial relationship with other structures. Moreover, 3D US can provide additional helpful information, such as the vascular connections, including feeding and draining vessels [37], and can indicate the potential for recanalization of the lesion after the first embolization confirmed by angiography. To illustrate the value of 3D US in diagnosis, we briefly describe a particular case which occurred in our department. In Figure 4A (left) the ultrasonography shows turbulent arterial flow deriving from the right uterine artery vessel into the anechoic area, consistent with a pseudoaneurysm and its supplying artery before the first embolization procedure. The subsequent angiography confirmed the diagnosis of oval pseudoaneurysm vascularized by the ectasic vessel originated by the right uterine artery. Since the woman desired future fertility, at that time an omolateral uterine right artery embolization with PVA particles and coils was performed. A new 3D-HD flow Doppler US evaluation four days later showed a still persistent blood flow deriving from a collateral branch of the left uterine artery vessel, revealing, an until then hidden, second blood source for the pseudoaneurysm (Figure 4B, right). Moreover, the continued blood loss from the patient led to repetition of the contralateral artery embolization. Subsequent angiography demonstrated evidence of the previous successful right uterine artery embolization and, further, showed a persistent feeding vessel belonging to the left uterine artery, which confirmed the 3D US diagnosis. A left uterine artery embolization was then performed, resolving the case successfully.

Detection of vascular malformation within the uterus is not straightforward and differential diagnosis must be carried out to consider the full range of possibilities. For instance, gestational trophoblastic neoplasia (GTN) tumor is another cause of vascular malformation due to arterio-venous shunts deriving from neoangiogenesis phenomena within the tumor mass. Differential diagnosis of myometrial nodules is represented by benign masses such as uterine fibroids, which usually appear as solid or partially colliquated masses, or focal adenomyosis, which can be ruled out by the lack of exaggerated vascularity in color Doppler flow imaging [38]. This is particularly important when assessing the patient without knowledge of BhCG testing to indicate requirement for the test. This is particularly as the US picture of vascular abnormality can persist after negativization of BHCG [38].

The risk of pseudoaneurysm rupture is proportional to its size and transmural pressure. Therefore, some pseudoaneurysms may spontaneously resolve through thrombosis [39]. In those few cases described, a spontaneous resolution might be the appropriate choice if pregnancy is considered in the near future [40]. It is worth noting that the literature has reported extensively on the sudden onset of life-threatening massive uterine hemorrhages from rupture of a pseudoaneurysm as the first symptom of UAP or as a recurrence after weeks of the presumed spontaneous resolution diagnosis of the lesion [41,42]. Pseudoaneurysm treatment methods include closing the fistula with a surgical or endovascular procedure [43]. Uterine embolization is currently the most used treatment since it preserves reproductive capacity, as widely described [44]. However, surgery is still considered the treatment of choice to be performed almost exclusively in the event of failure of embolization or hemodynamic instability [45]. Uterine embolization is a safe and efficient treatment with 93% effectiveness and a 0.4% complication rate [43].

Van den Haak [46], in a review focusing on the use of manipulators in laparoscopic surgeries, reports that many authors mention the importance of the cephalic movement of the uterus to prevent damage to the urinary tract since this increases the distance between the ureter and the cervix. Kavallaris et al. [47] reported a rate of 0.5–1.0% of ureteric injuries in cases of TLH (total laparoscopic hysterectomy). However, there is still little evidence in the literature concerning the efficacy and safety of uterine manipulators [46].

Despite everything, nowadays, the uterine manipulator is considered an essential tool during total laparoscopic hysterectomy (TLH) surgery or in complicated surgery, such as endometriosis or oncology. Many factors are relevant during application of the procedure. For instance, the wider the angular extent of the manipulator, the better is the exposure of the uterine walls and ligaments. Lateral movement allows exposure of the pelvic ligaments, and utero-ovarian and anterior-posterior leaves of the broad ligament. Antero-posterior movement exposes the anterior uterine wall if associated with elevation movement of the vesico-uterine fold, and posterior uterine wall [48]. For example, during rectovaginal endometriosis surgery, the manipulator is particularly important because the upward motion of the uterus into the pelvis enhances the view of the uterosacral ligaments and cul-de-sac [15].

The newer uterine manipulators are designed to address the effects of obesity and other anatomic impediments to the uterine flexion range. Some of the best-known instrument models are Clermont-Ferrand, Hohl, Endopath, RUMI, Vcare, Dr. Mengeshikar, Clearview, Vectec, Valtchev, and the McCartney tube.

The choice of the best manipulator depends on the operating surgeon’s needs according to the type of surgery. The Hohl manipulator has a 130° range of motion in the anterior-posterior plane and is a reusable model, straightforward to use mainly in TLH, where it provides an excellent elevation to the uterus; moreover, it can be utilized during advanced procedures, such as endometriosis [15]. The Clermont-Ferrand is a reusable manipulator and provides a 140° movement of the uterus. It provides good delineation of the vaginal fornices and maintains the pneumoperitoneum after the colpotomy incision. This instrument requires cervical dilatation before the insertion, so it cannot be used in cases of cervical stenosis (Karl Storz Co) [49]. The Vcare uterine manipulator is disposable, allows an extensive range of motion, offers a good presentation of the vaginal fornices at colpotomy incision, and maintains the pneumoperitoneum [46]. The RUMI manipulator system consists of the following four components: the RUMI uterine tip, where the balloon is fixed, the Koh cervical cup (the vaginal fornices delineator), the Koh colpopneumo-occluder, and the RUMI handle [50]. This device helps dissect the cervix and vagina during laparoscopic surgery because the movement lengthens the uterosacral ligaments allowing an accurate delineation of the cervical-vaginal region. This system is helpful in laparoscopic hysterectomy in patients with enlarged uteri and endometriosis surgery [15]. The Dr. Mangeshikar uterine manipulator has been used in TLH and endometriosis surgery; its main advantage is ease of assembly and use. For instance, during a THL, the manipulator allows for good stretching and a wide range of movement in several directions. Its good mobility facilitates dissection of the uterine arteries, decreasing the risk of ureteric injury. Moreover, after this kind of surgery, the uterus can be delivered vaginally, maintaining the entire assembly [51].

Despite all the recommendations of using manipulators in TLH and other complex surgeries, it is not rare to find complications caused by these instruments. Wu et al. [52] reported two cases of iatrogenic uterine rupture due to hyperinflation of the balloon of the RUMI manipulator in laparoscopic tubal surgery. During the chromoperturbation, the methylene blue solution was wrongly injected into the inflation port, resulting in a massive hematoma. Ali and Teskin [53] reported a case of posterior uterine fundus perforation and intestine penetration during a laparoscopic exploration caused by the tip of the Hohl manipulator. In other findings, some reports advise against the employment of some models. It is reported (for instance, see [46]) that the Clearview model, although easy to use, has features (the absence of a cervical cup), which prevents pneumoperitoneum and is therefore not recommended for TLH. Equally easy to use is Vcare; however, it is too light in a larger uterus, and disintegration of the instrument, or some parts being left behind, has been reported [46].

The choice of the best manipulator depends on the operating surgeon’s needs according to the type of surgery. We summarize the main features, indications, advantages, and disadvantages of the manipulators in Table 1. Mettler and Nikam [15] considered that the TLH-Dr Mangeshikar model and the Clermont-Ferrand model are suitable in advanced cases of TLH and endometriosis (due to excellent capabilities and versatility), especially when the recto-vaginal area is involved. RUMI and Hohl are also recognized as suitable for endometriosis procedures.

Zygouris et al. [55] presented only one case of ureteral injury and three cases of bladder opening during laparoscopy in a retrospective study with 1023 patients who underwent laparoscopic hysterectomy without the use of a manipulator. The authors concluded that a TLH without using a manipulator is achievable and safe when performed by an experienced laparoscopic surgeon, with short surgical time and a low rate of complications. Khalek et al. [54], in a narrative review, described other complications from the use of manipulators, such as uterine rupture due to over-inflation of the manipulator balloon (using Hohl, V-care, RUMI, and Clearview manipulators), bowel perforation, and uterine rupture with the use of Hohl manipulator or lacerations of the vagina with excessive bleeding (V-care and RUMI). We emphasize the importance of correct use and caution for the manipulation. Nevertheless, the authors have highlighted no qualified data/studies describing ureteric and bladder injury rates during TLH with or without manipulators.

## 4. Conclusions

When analyzing the literature, we found that most of the studies looking at internal injuries caused by manipulators in gynecological procedures refer to “external” or “massive” damage to the uterus (or adjacent organs). In other words, the instrument itself could cause complications, such as uterine rupture, bowel penetration, hemorrhage from laceration, or retroperitoneal hematoma caused by uterine perforation. The most appropriate choice, based on the particular characteristics of each manipulator in relation to the type of surgery to be programmed, can perhaps help prevent potential damage to the uterus and contiguous organs and consequently must be managed with care. From a different perspective, this report described a case where the manipulator was the agent that caused the UAP, which was a kind of injury virtually impossible to detect during (or just after) the procedure, mainly because of the asymptomatic damage. Therefore, it is of the utmost importance to rapidly follow-up patients (preferably using the US) who underwent surgery with a uterine manipulator and consider the possibility of complications, such as UAP, especially if there is a clinical condition of blood loss. In this context, the ultrasound and color Doppler features typical of the lesion alone can provide a precise diagnosis without needing the most expensive and sophisticated radiological techniques but considering confirmatory angiography due to embolization therapy. Three-dimensional US may further clarify the typical and modified spatial relationship of the vessel tree.

## Figures and Tables

**Figure 1 diagnostics-12-00164-f001:**
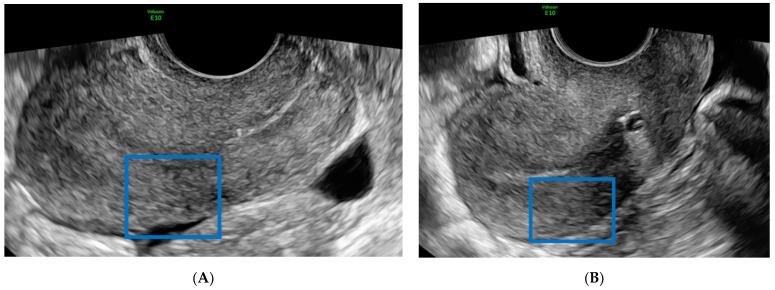
(**A**) Preoperative ultrasonography of the pelvis (the blue square denotes the area of interest). (**B**) Postoperative ultrasonography of the pelvis.

**Figure 2 diagnostics-12-00164-f002:**
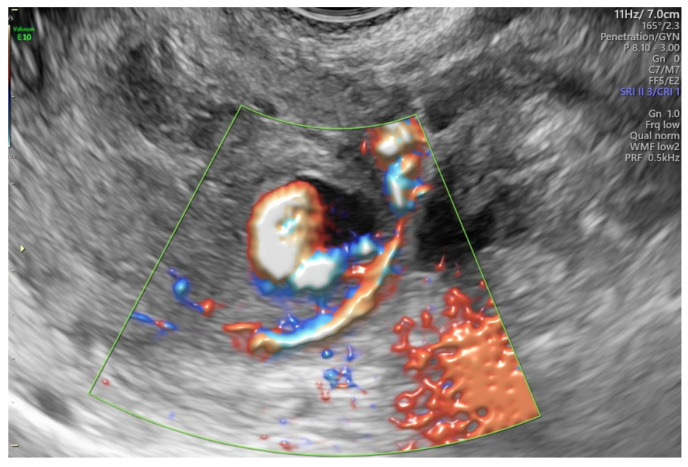
Acquired pseudoaneurysm from the ascending branch of the left uterine artery. Note the to-and-fro sign and the yin-yang image.

**Figure 3 diagnostics-12-00164-f003:**
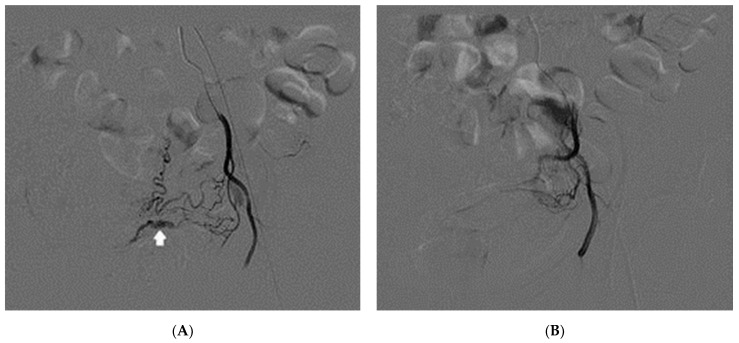
(**A**) Selective angiography of the left internal iliac artery before embolization: the contrast media injection shows an abnormal dilatation with contrast media extravasation (indicated by the arrow) out of the distal branch of the left uterine artery. (**B**) Angiography after embolization with embolic particles; the lesion is no longer visible.

**Figure 4 diagnostics-12-00164-f004:**
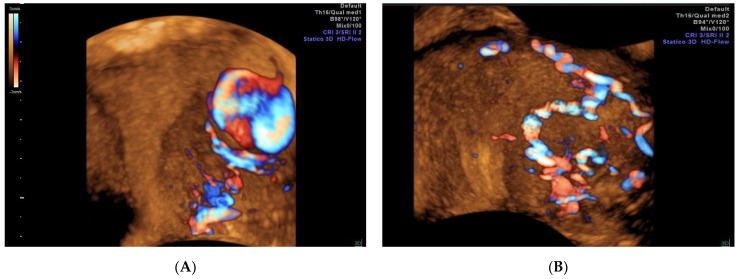
(**A**) 3D-HD flow color Doppler rendering image showing turbulent arterial flow into the anechoic area deriving from the right uterine artery vessel. (**B**) 3D-HD flow rendering image after the first embolization procedure showing the persistence of the flow deriving from a collateral branch of the left uterine artery vessel.

**Table 1 diagnostics-12-00164-t001:** Summary of the main features, indications, advantages, and disadvantages of the manipulators.

Manipulator	Characteristics	Employment	Advantages	Disadvantages
Hohl [15,53,54]	Traumatic Reusable Pneumoperitoneum	(T)LH	Movement range Independent movements Less traumatic	Restriction in the elevation of the uterus May cause cervical bleeding or uterine rupture
Clermont-Ferrand [15,49]	Traumatic Reusable Pneumoperitoneum	(T)LH Endometriosis of cul-de-sac	Movement range Independent movements Allows easy grasping of the uterine pedicles and lateral fornices	Requires dilatation of the cervix Requires specialized training Complex to assembly Expensive
Vcare [46]	Not reusable Pneumoperitoneum	(T)LH	Good presentation of the vaginal fornices Independent movements Good handling	Disposable instrument Too light when dealing with big uterus May leave behind parts of the manipulator inside the patient May cause laceration of the vagina
RUMI ^a^ [15,50,52,54]	Traumatic Partially reusable Pneumoperitoneum	(T)LH	Movement range Good delineation of the vaginal fornices Good when using coupled to US	Hard to handle (particularly in the case of narrow vagina) Difficult to assembly Restricted elevation of the uterus May cause laceration of the vagina May leave behind parts of the manipulator inside the patient
Clearview [46,54]	Traumatic Not reusable Not maintain the pneumoperitoneum	All procedures except (T)LH	Largest movement range Easy to handle and assemble Allows single handling by the surgeon (no need of assistant)	No delineation of the vaginal fornices May cause uterine perforation May leave behind parts of the manipulator inside the patient
Dr Mangeshikar [15,51]	Traumatic Reusable Pneumoperitoneum	(T)LH Endometriosis of cul-de-sac	Low cost Wide range of motion Good presentation of the vaginal fornices Easy to handle and assemble Low risk of ureteric injuries	

^a^ refers to the RUMI system with the KOH colpotomizer.

## Data Availability

Data sharing is not applicable to this article.

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
