# Peer review of "Ultrasound Imaging of Acquired Myometrial Pseudoaneurysm: The Role of Manipulators as an Unusual Cause during Laparoscopic Surgery"

_diagnostics, 2022, doi:10.3390/diagnostics12010164_

Round 1
Reviewer 1 Report
An interesting case report, with a useful evaluation of indications, advantages, and disadvantages of laparoscopic surgeries manipulators.
Author Response
We appreciate a lot the kind compliments concerning our work and for the time spent to analyze this manuscript!
We have checked the spelling of the text!
Reviewer 2 Report
This Ultrasound Imaging paper describes a case report as an unusual cause of Myometrial Pseudoaneurysm Acquired during laparoscopic surgery and due to role of manipulators. The case is interesting and sound, deserving publication after having resolved few issues:
Did the patient undergo preoperative ultrasound? it would be beneficial to add the pictures.
Did the patient undergo beta-hCG testing preoperatively, please add. This does not diminishes the importance of discussing differential diagnosis for readers, as exposed below.
An intraoperative picture of the region of interest may be added, it would be beneficial showing the area in which the lesion developed.
Detection of vascular malformation within the uterus requires some clarifications. It is clear that a differential diagnosis must be carried out and other causes of must be considered. Gestational trophoblastic tumor is another relatively common cause of vascular malformation and this must be cited briefly in the discussion and appropriate reference added (ref 1). Arterio-venous shunts in the case of GTN are due to neoangiogenesis phenomena within the tumor mass are also common. Differential diagnosis of myometrial nodules is represented by benign masses such as adenomyosis and uterine fibroids, which can be ruled out by the lack of exaggerated vascularity at Color Doppler flow imaging. This is important when assessing the patient without the knowledge of BhCG testing in order to indicate the test. More so the US picture of vascular abnormality can persist after negativization of BHCG. (ref 1)
Other relevant citations must be considered. Uterine artery pseudoaneurism should be considered in pregnant women with severe abdominal and pelvic pain, once other obstetrical factors have been excluded. DIE might be the underlying diagnosis. It is a rare but potentially life-threatening condition for mother and fetus. (2)
Uterine artery pseudoaneurism is also rare complication after Cesarean scar pregnancy treatment that can lead to fatal massive hemorrhage. (ref 3)
The role of 3D ultrasound is carried forward in the conclusion. However, I do not see any picture or data concerning this technique. Please add or erase the sentence.
The authors state that this is the second case found in the literature. Please add citation to the other case troughout the text (line162).
Consider the fact that endometriosis treatment with laser ablation rather than excisional surgery often required reduced tissue traction and potentially reduced tissue damage. May this lead to reduced risk of such iatrogenic insults? Please also consider this issue. What are the suggestions of the authors in order to prevent future occurrence of these iatrogenic damages? This conclusion may be included in the conclusion of the main text and abstract.
Minor
“Therefore” written twice at line 318.
Doppler must be written in capital letter throughout the manuscript (e.g. line 321)
The authors should be careful with text editing (lines 287-293)
References
- Cavoretto P, et al. A Pictorial Ultrasound Essay of Gestational Trophoblastic Disease. J Ultrasound Med. 2020 Mar;39(3):597-613. doi: 10.1002/jum.15119. Epub 2019 Aug 29. PMID: 31468566.
- Zwimpfer TA, et al. Uterine pseudoaneurysm on the basis of deep infiltrating endometriosis during pregnancy-a case report. BMC Pregnancy Childbirth. 2021 Apr 9;21(1):282. doi: 10.1186/s12884-021-03753-1. PMID: 33836672; PMCID: PMC8034083.
- Wang J, et al. Uterine artery pseudoaneurysm after treatment of cesarean scar pregnancy: a case report. BMC Pregnancy Childbirth. 2021 Oct 9;21(1):689. doi: 10.1186/s12884-021-04166-w. PMID: 34627190; PMCID: PMC8501730.
- Candiani M, et al Assessment of ovarian reserve after cystectomy versus 'one-step' laser vaporization in the treatment of ovarian endometrioma: a small randomized clinical trial. Hum Reprod. 2018 Dec 1;33(12):2205-2211. doi: 10.1093/humrep/dey305. PMID: 30299482; PMCID: PMC6238368.
Author Response
We thank reviewers for their constructive criticism, and time spent to analyze this manuscript. The responses, and explanations related to their comments are listed in the following:
- As a requisition of further images of preoperative US and an intraoperative picture of the region of interest, we add on a revised version of the manuscript a two new figures highlighting these points;
- As a response to the reviewer’s comment, we also add on the text of the new version the information about negative preoperatively BhCG result for the patient;
- As a response to the reviewer’s comment concerning differential diagnosis, we add a paragraph (and the reference suggested by the reviewer) in the Discussion section as follows: "Detection of vascular malformation within the uterus is not straighforward and differential diagnosis must be carried out to consider all the range of possibilities. For instance, Gestational trophoblastic neoplasia (GTN) tumor is another cause of vascular malformation due to arterio-venous shunts deriving from neoangiogenesis phenomena within the tumor mass. Differential diagnosis of myometrial nodules is represented by benign masses such as uterine fibroids, which usually appear as solid or partially colliquated masses, or focal adenomyosis, which can be ruled out by the lack of exaggerated vascularity at Color Doppler flow imaging [ref 1]. This is particularly important when assessing the patient without the knowledge of BhCG testing in order to indicate the test. More so the US picture of vascular abnormality can persist after negativization of BHCG [ref 1]."
- As a response to the reviewer’s comment related to obstetric cases, we emphasized that the focus of this present work concerns only the gynecological frame, thus we passed over any obstetric event concerning the subject. Even if we do agree with the reviewer that the cited cases are relevant, we limited our analysis on the frame of this case report. Nevertheless, we add a sentence on the introduction talking about the pseudoaneurysm issue also during/after pregnancy citing the two suggested references (ref. 3 and 4);
- As suggested by the reviewer, we add on the revised manuscript 2 extra figures on 3D US Color Doppler images and the following paragraph in the Discussion section: "In order to illustrate the 3D US ability in diagnose, we briefly describe a particular case occurred in our department. In Figure 4A (left) the ultrasonography shows a turbulent arterial flow deriving from the right uterine artery vessel into the anechoic area consistent with a pseudoaneurysm and its supplying artery before the first embolization procedure. The subsequent Angiography confirmed the diagnosis of oval pseudoaneurysm vascularized by the ectasic vessel originated by the right uterine artery. Since the woman desired future fertility, at that time an omolateral uterine right artery embolization with PVA particles and coils was performed. A new 3D-HD flow Doppler US evaluation four days later showed a still persistent blood flow deriving from a collateral branch of the left uterine artery vessel, revealing an until then hidden second blood source for the psedoaneurysm (Figure 4B, right). Moreover, the continued blood loss from the patient led to repeat the contralateral artery embolization. In fact subsequent angiography demonstrated evidence of the previous successful right uterine artery embolization and, further, it also showed a persistent feeding vessel belonging to the left uterine artery, which confirmed the 3D US diagnosis. A left uterine artery embolization was then performed getting the case resolving successfully."
- We have explicitly added the reference of the first similar case found in literature, as requested by the reviewer;
- Concerning the question made by the reviewer, "What are the suggestions of the authors in order to prevent future occurrence of these iatrogenic damages? " we add on the Conclusion section the sentence: "The most appropriate choice based on the particular characteristics of each manipulator in relation to the type of surgery to be programmed, can perhaps help prevent potential damage to the uterus and contiguous organs and consequently must be managed with care."
- Concerning the reviewer's comment on endometriosis treatment with laser ablation and particularly the suggested citation (ref. 4), from our point of view the discussion of the article is beyond the scope of our work, since it is related to the ovarian damage while we talk about the damage to the uterus from a manipulator.
- Finally, we checked the typos of the text, but we highlight that there was no correspondence between the lines indicated by the reviewer with the lines inside the manuscript file.
Reviewer 3 Report
There are some minor English spelling checks needed, on line 318 there is a repeated word “Therefore”.
Author Response
We thank reviewers for the time spent to analyze this manuscript. We have checked the spelling of the whole text! However, there is no accurate correspondence between the line indicated by the reviewer and that one on the manuscript file.